# Early Association Factors for Depression Symptoms in Pregnancy: A Comparison between Spanish Women Spontaneously Gestation and with Assisted Reproduction Techniques

**DOI:** 10.3390/jcm10235672

**Published:** 2021-11-30

**Authors:** David Ramiro-Cortijo, Cristina Soto-Balbuena, María F. Rodríguez-Muñoz

**Affiliations:** 1Department of Physiology, Faculty of Medicine, Universidad Autónoma de Madrid, C/Arzobispo Morcillo 2, 28029 Madrid, Spain; david.ramiro@uam.es; 2Department of Obstetrics and Gynecology, Hospital Universitario Central de Asturias, Avenida Roma s/n, 33011 Oviedo, Spain; cristsobal@yahoo.es; 3Department of Psychology, Faculty of Psychology, Universidad Nacional de Educación a Distancia, C/Juan del Rosal, 10, 28040 Madrid, Spain

**Keywords:** depression symptomatology, pregnancy, risk factors, protective factors, assisted reproduction techniques

## Abstract

Women with assisted reproduction techniques (ART) have a different psychological profile than women with a spontaneous pregnancy. These differences may put the former group at higher risk for depressive symptomatology. Our aim was to determine what sociodemographic factors and psychological variables interact with early depressive symptoms in pregnant women with ART. This is a cross-sectional, non-interventional, and observational study where a total of 324 women were analyzed in the first trimester of pregnancy at the Hospital Universitario Central de Asturias (Spain). Women completed a sociodemographic questionnaire, the Patient Health Questionnaire (PHQ-9), the Generalized Anxiety Disorder 7-item Scale, the Resilience inventory, the General concerns (ad hoc scale), the Stressful life events, and the prenatal version of Postpartum Depression Predictors Inventory-Revised (PDPI-R), including socioeconomic status, pregnancy intendedness, self-esteem, partner support, family support, friends support, marital satisfaction, and life stress. According to our models, women undergoing ART had significantly increased the PHQ-9 scores (β = 6.75 ± 0.74; *p*-value < 0.001). Being single also increased the PHQ-9 score. Related to the psychological variables, anxiety (β = 0.43 ± 0.06; *p*-value < 0.001) and stressful life events (β = 0.17 ± 0.06; *p*-value = 0.003) increased PHQ-9 scores. In contrast, resilience (β = −0.05 ± 0.02; *p*-value = 0.004), self-esteem (β = −1.21 ± 0.61; *p*-value = 0.048), and partner support (β = −1.50 ± 0.60; *p*-value = 0.013) decreased PHQ-9 scores. We concluded that women undergoing ART need interventions to reduce anxiety and stressful life events, and to improve resilience, self-esteem, and emotional partner support to prevent depressive symptomatology during this important phase in their lives.

## 1. Introduction

In industrialized countries, the age of first pregnancy is rising due to social and economic changes [1]. Pregnant women who are older may be psychologically and socially better prepared to handle maternity than younger mothers due to having greater education [2]. However, older women may have fertility issues that require the use of assisted reproduction techniques (ART) to become pregnant [3]. Overall, the trend is an increase of the use ART to address the higher infertility rates among women in industrialized countries [4,5]. For example, in Spain, there were 12,000 neonates by ART in 2009 [6] and 37,094 infants in 2018 [7]. Given the additional medical interventions, women with an ART-mediated pregnancy may exhibit a different psychological pattern than women with a spontaneous pregnancy [8]. A review suggested that women who become pregnant by ART seem to have an increased risk of depression in her life [9]. However, there is controversy regarding pregnant women by ART and antenatal depression [10]. Although there is no postulated model that explains why depression during pregnancy could be associated with ART, there are described factors that could induce it in these women. It seems that the determinant point that associates antenatal depression and ART would be the time of pregnancy when depression was assayed, maternal age at the beginning of fertility treatment as well as years of infertility and experience of miscarriage [11,12,13].

Research indicates the negative role of maternal depression during pregnancy on fetal and infant development [14,15]. Up to 15% of women experience high levels of depressive symptoms during pregnancy [14]. Risk factors associated with depressive symptoms include unemployment, low educational level, unplanned pregnancy, history of depression, low partner and social support, smoking, and negative life events [16,17,18]. The different attitude of ART conception to coping with the pregnancy could modulate the depression symptomatology. Most of the women who conceive ART have experienced several years of infertility, and infertility has been associated with higher rates of depression [19,20]. However, most of these studies were performed in the late stage of pregnancy or even during the postpartum period. Therefore, additional research is needed to examine if depression is detected earlier in gestation among women with ART pregnancies, and its associations with risk and protective factors.

In addition to depression, anxiety is also highly prevalent during pregnancy among women with ART pregnancies [21]. However, there is more limited and mixed research regarding the protective and risk factors associated with anxiety for this population. A review shows that ART pregnancies had greater pregnancy-specific anxiety and poorer quality of life than spontaneous gestations [22]. However, another review shows that women with ART pregnancies could have more a positive attitude toward pregnancy demands, higher levels of maternal-fetal attachment, and even less depressive symptomatology than women with natural conception pregnancies [23]. The methodological limitations to analyzing anxiety and differences across sociodemographic variables could explain the inconsistencies in results related to the impact of ART on maternal psychological status.

Anxiety, general concerns, and stressful life events during pregnancy could be precipitant factors associated with higher levels of depressive symptomatology. According to the most influential theory of stress and coping development by Lazarus and Folkman [24], when the women cannot respond to a potentially stressful situation judged as a threat, negative emotional consequences happen [25]. During pregnancy, this health framework suggests that multiple psychosocial variables synergically interact on maternal mental health towards a more depressive symptomatology in the absence of optimal coping strategies. There is evidence suggesting that a woman’s exposure to stressful life events during pregnancy is linked to multiple adverse outcomes [26]. However, the stress life events could have ambiguous information about the type of stress measured during the prenatal period. Life events such as the death of a relative, serious financial issues, unemployment, or exposure to a natural disaster could generate more anxiety and psychological distress than others such as alcoholism/drug addition of the partner, and relationship difficulties [27]. However, through life, the first events may have a lower prevalence than the second ones, acting differently on maternal depressive symptomatology. The ART could be considered as a stressful event for women, leading to poor obstetrical outcomes [28]. In contrast, women who perceive satisfactory social support (partner, family, and friends) also perceive higher capacities and greater self-efficacy, evaluating the situation as less stressful [29]. Furthermore, there are data to support that resilience has a positive role in maintaining positive emotion by the end of gestation [30]. Therefore, it seems essential to determine early in the pregnancy those variables that may be protective and risk factors of developing depressive symptomatology. Although they have been poorly explored in women with ART, they are important form the point of view of psychological vulnerability.

The purpose of the study, as the primary outcome, was to estimate what sociodemographic factors and psychological variables could impact on early depression symptoms in pregnant women undergoing ART. To our knowledge, there are no known studies that contrast the prevalence and risk factors for antenatal depression of pregnant women derived of ART. In this study, ART pregnant women will be compared to spontaneously. The first aim was to estimate in the early of gestation, the scores of depression symptomatology in women derived of ART. Secondly, to evaluate the sociodemographic and psychosocial risk and protective factors associated with antenatal depression for each group. Finally, to estimate the extent to which the sociodemographic and psychological factors can predict the development of antenatal depression, both in general (ART and non-ART), and for the subset ART group. Exploring this area could establish comprehensive screening and improve prevention policies in obstetrics, considering the rising use of ART and the relevance of depression in pregnancy to subsequent maternal–neonatal health.

## 2. Materials and Methods

### 2.1. Cohort and Study Design

This cross-sectional, non-interventional and observational study was approved by the Hospital Universitario Central de Asturias (HUCA, Oviedo, Asturias; Spain) Research Ethics Committee (Refs. 18/18). The sample size was estimated by considering that up to 15% of women would develop antenatal depression symptomatology [14] with an error margin of 5%. The required sample size was 196 women.

Women treated at the Obstetrics and Gynecology Service from HUCA from 2014 to 2015 were invited to participate in the study. HUCA is a public hospital located in the North of Spain, which serves a mostly urban setting but also the rural areas close to it. All single pregnant women who had the HUCA as their obstetric reference unit to control their pregnancy were asked to participate in the study in her first obstetrical visit (9th week of gestation). A total of 679 women were contacted during their first obstetrical visit. Women entered in the study confidentially and voluntarily signed an informed consent. A total of 400 out 679 women wanted to be included in the study with a correct understanding of Spanish language. In addition, those women who matched the inclusion and exclusion criteria were selected. Inclusion criteria included being pregnant and receiving prenatal services at the hospital. To avoid potential confounding factors related to depression symptomatology that could be interfering with our purpose, we used exclusion criteria which were women with previous mood or depression disease diagnoses [31] and risk factors for comorbidities (smoking habits and regular alcohol intake [32]). The final cohort to be analyzed in the study was 324 women (Figure 1). Considering this as an observational and cross-sectional study, increasing the sample size versus the estimated one could have benefits in suppressing the potential risk of bias. In addition, STROBE Statement for reporting cross-sectional studies was followed [33] (see checklist in Appendix A).

Participants completed a set of questionnaires including sociodemographic and psychological variables during the first trimester.

**Sociodemographic Variables.** These data were collected by an ad hoc questionnaire, including maternal age (years), marital status (married/separated/single), education level (primary degree/high school degree/university degree), country of birth (Spain/Europe/Latin America/Other), time in Spain (years) and place origin (in non-Spanish women), and employment (yes/no).

**Medical Data.** These variables were self-reported and recorded including previous comorbidities (yes/no, such as hypertension, diabetes mellitus, cancer, asthma, and immunological diseases among others), number of pregnancies, number of children alive, previous abortions and C-sections, and type of reproduction (spontaneous/ART).

### 2.2. Maternal Psychological Instruments during Early Pregnancy

Women answered psychological applications in the 9–11 week of gestation (first trimester of pregnancy) during a programmed visit to the hospital. Psychological instruments are described below:

**Patient Health Questionnaire (PHQ-9)**. The PHQ-9 [34] is an instrument that measure the specific characteristics of major depression disorder according to DSM-IV. Furthermore, the DSM-5 Work Group for Depressive Disorders considers the PHQ-9 as the preferred measure for depression [35]. The PHQ-9 has been used and validated in Spanish cohorts [36,37,38]. This is a 9-items instrument with a four-level Likert scale with responses from 0 (never) to 3 (almost every day). A higher PHQ-9 score indicates higher severity of depressive symptoms (range 0–27). The cut-off recommended for risk of depression would be ≥10 [39,40]. Previous Spanish studies have shown an internal consistency of 0.81 in the PHQ-9 score [38]. In our cohort, the internal consistency was 0.81.

**Generalized Anxiety Disorder 7-Item Scale (GAD-7)**. The GAD-7 [41] measures anxiety symptoms. According to a review of pregnancy, the GAD-7 showed superior psychometric results compared with other instruments [42]. The GAD-7 has been used and validated in Spanish cohorts [43,44]. This is a 7-items instrument with a four-level Likert scale with responses from 0 (not at all) to 3 (nearly every day). A higher GAD-7 score indicates higher anxiety symptoms (range 0–21) [41]. The GAD-7 has been reported an internal consistency of 0.89 in the first trimester of pregnancy [21]. In our cohort, the internal consistency was 0.89.

**Resilience Inventory**. The Resilience inventory [45] assess the seven dimensions of resilience (including: positive attitude, sense of humor, perseverance, religiosity, self-efficacy, optimism, and goal orientation) with 20 items. This scale measures only dispositions to resilience, without testing the risks and competencies, according to the theory of the resilience [46]. This scale has a five-level Likert scale with responses from 1 (not at all) to 5 (always). Examples of questions: “I see the positive side of life” and “I accept that problems are part of life”. Previously, the scale obtained an internal consistency of 0.89–0.93 [47]. In our cohort, the internal consistency was 0.92.

**General Concerns**. It is an ad hoc scale, which was elaborated according to the phases to build of psychological scales [48]. This scale assesses the concerns derived from the pregnancy process. There are some adaptations of general concern scaled to pregnancy [49]. However, we used a variation, consisting of three questions focused on the child’s health, delivery, and maternity feelings, adapted to the current study cohort. This test had a response scale from 0 (none) to 4 (very). Items included in the scale to determine pregnancy worries were: “how much does it worry you that: (1) the newborn is born well? (2) the delivery works? and (3) how will I be as a mother?”. Our general concerns scale had an internal consistency of 0.74.

**Stressful Life Events (SLE)**. This scale is a checklist in which different stressful events (e.g., a personal or a family member’s illness, financial problems, alcoholism or drug addiction of the partner, possible mistreatment by the partner, and relationship difficulties) usually described in the literature are evaluated [50]. This scale has been published in Spanish for pregnant women [21]. It assesses the occurrence of the 12 events within the last six months and the degree of stress produced with a five-level Likert scale with responses from 0 (the event was not present) to 4 (it produced great stress). Higher stressful life events score showing higher stress and discomfort in life (range 0–48). Previous studies endorse the adequacy psychometric properties [21,51]. In our cohort, the internal consistency was 0.62.

**Postpartum Depression Predictors Inventory-Revised (PDPI-R)**. The PDPI-R [52] includes 13 risk factors related to the development of postpartum depression. This scale determines whether risk factors could be present in a woman [53]. The first 10 predictors comprise the prenatal version of the PDPI-R, including prenatal depression (not included in this study), prenatal anxiety (not included in this study), prior depression (not included in this study), marital status (not included in this study), socioeconomic status (SES, 1 item; low/medium/high), pregnancy intendedness (1 item; no = 0/yes = 1; range 0–1), self-esteem (3 items; no = 0/yes = 1; range 0–3), social support (including: 4 items for partner support, 4 items for family support, and 4 items for friends support; no = 0/yes = 1; range 0–4 each dimension), marital satisfaction (3 items; no = 0/yes = 1; range 0–3), and life stress (7 items; no = 0/yes = 1; range 0–7). The prenatal Spanish version of PDPI-R was validated and used [54,55,56]. Higher scores correlate to more risk of postpartum depression [53]. Previous studies reported an internal consistency of this index used among women during pregnancy as 0.86 [54,55]. In our cohort, the internal consistency was 0.80.

### 2.3. Statistical Analysis

The data were summarized as a median and interquartile range [Q1; Q3] in quantitative variables and relative frequency (%) and sample size (*n*) in qualitative variables. The Mann–Whitney’s U test was performed to study the differences between two groups and Fisher’s exact test was used to check the association of categories, when association was significant, odd ratio (OR), and 95% confidence intervals [95% CI] were reported. The Spearmen’s rho (ρ) coefficient was used to test the correlations between quantitative variables, and when the psychological variables showed a dichotomous dimension, the numerical codification was applied.

To test the contribution of ART controlled by sociodemographic and psychological variables to the maternal depression score (PHQ-9) as the dependent variable, the general regression models were built. Three models were applied: (1) unadjusted ART (by itself), (2) ART adjusted by sociodemographic characteristics with known importance in developing depression and related to ART as the covariables, and (3) ART adjusted by psychological covariables, which could be a risk and protective factor for depression. In the second model, the variables with more than two categories were clustered in two categories according to the risk factor for depression as marital status (married/single; married was the reference category), education level (non-university degree/university degree; university degree was the reference category), and country of birth (Spain/not Spain; Spain was the reference category). Furthermore, the model 3 was also adjusted for the significant variables of the model 2. In a secondary analysis, we applied the model in a subset of the cohort considering only women derived of ART. In all models, the standardized beta coefficients (β), standard error, and *p*-value associated were extracted. The adjusted R-squared (Adj. R^2^) for each model was also reported.

Overall, the statistical significance was established as *p*-value < 0.05. All data analyses were performed using R software (version 4.1.1, R Core Team 2021, Vienna, Austria) within RStudio (version 1.1.453, Rstudio, Inc., Vienna, Austria) using the rio, dplyr, Hmisc, rsq, epitools, devtools, and compareGroups packages.

## 3. Results

In the study population, the maternal age was 34.0 [30.0; 36.0] years old. A 90.1% (292/324) were Spanish women, the women whose country of birth was not Spain had lived in Spain, in median, 10 years (Min = 1 years; Max = 38 years). A 52.5% (170/324) of the women had a university degree, the 45.7% (148/324) had a primary school degree (up to age 12). The 73.7% (238/323) were employed and 65.4% (212/323) were married. A 16.7% (54/323) of the cohort showed comorbidities, and 28.8% (92/319) used medications regularly. Primiparous women were a 47.2% (153/324) of the cohort, 38.5% (124/322) had one child, and 3.7% had more than one; 18.8% (57/303) of the women had an abortion, and 7.5% (24/321) had a C-section before. Women who conceived via ART represented 9.3% (30/321), and 93.3% of them (28/30) did so via in vitro fecundation.

### 3.1. Sociodemographic Characteristics Related to ART and Depression Scale

The sociodemographic characteristics of women who experienced spontaneous conception (non-ART) and ART were described in Table 1. The mothers with ART were significantly older than non-ART mothers. In addition, there was more prevalence of first maternity in the ART than in the non-ART group. Marital status, level of education, country of birth, employment, comorbidities, and previous abortions and C-section were not significant between groups (Table 1).

Overall, PHQ-9 shows a score of 5.0 [3.0; 8.0]. There were no statistical differences in PHQ-9 score between women with ART and non-ART (non-ART = 5.0 [3.0; 8.0], ART = 6.0 [3.0; 11.0]; *p*-value = 0.196). Considering the cut-off of ≥10 to detect women with potential depression symptoms, the PHQ-9 detected a 15.9% (49/309) of the cohort. The women with PHQ-9 score ≥ 10 were in non-ART 13.6% (40/294), and those in the ART group were 32.1% (9/28). The PHQ-9 score ≥ 10 was associated with type of reproduction (OR = 2.8, 95% CI = [1.2; 6.7]; χ^2^ = 4.72; *p*-value = 0.030).

The relationship between sociodemographic characteristics and depression scale is shown in Table 2. There were no significant variables. The PHQ-9 score ≥ 10 shows a non-significant trend in relation to educational level (Table 2).

### 3.2. Behavior of Psychological Variables with Depression Scale

Table 3 represents the correlations between scores of psychological instruments and the depression scale explored in the full cohort. This analysis guides us to choose what the psychological variables are the candidates to be included in the subsequent analysis. The PHQ-9 revealed significant and positive correlations with anxiety (GAD-7), stressful life events (SLE), general concerns, and life stressors and negative correlations with resilience, self-esteem, social support (in its three dimensions: partner, family, and friend support), and socioeconomic status related to PDPI-R. The marital satisfaction and pregnancy intendedness were not correlated with the explored depression scale (Table 3) and will not be introduced in the following models.

### 3.3. Depression Scale and ART Models Controlled by Demographic and Psychological Variables

According to our generalized models, ART significantly increased the PHQ-9 scores by itself (β = 6.75 ± 0.74; *p*-value < 0.001) and even adjusted by sociodemographic characteristics and psychological variables (Table 4). Marital status also increased the PHQ-9 scores, being single women constituting a risk factor for depression (β = 1.14 ± 0.50; *p*-value = 0.022). Related to the psychological variables, anxiety (GAD-7: β = 0.43 ± 0.06; *p*-value < 0.001) and stressful life events (β = 0.17 ± 0.06; *p*-value = 0.003), significantly increased the PHQ-9 score. Conversely, the resilience (β = −0.05 ± 0.02; *p*-value = 0.004), self-esteem (β = −1.21 ± 0.61; *p*-value = 0.048), and partner support (β = −1.50 ± 0.60; *p*-value = 0.013) categories of postpartum depression predictor inventory (PDPI-R) were protective factors in reducing PHQ-9 score (Table 4).

### 3.4. Depression Scale in a Subset of Women with ART

We applied the model to the subset of the cohort of pregnant women derived from ART. This model was controlled by marital status (significant sociodemographic variable in model 2) and psychological variables that were significant in Table 3. When the model was only applied to the ART subset, none of the variables were associated with PHQ-9 scores (Table 5). Women with medicalized pregnancies could present qualitative differences in the psychological pattern when compared jointly with women with spontaneous pregnancies and considering the time of gestation to assay the variables.

## 4. Discussion

In this study, we found that in early pregnancy, scores of the PHQ-9 were associated with pregnant women derived by ART. While there were no significant psychological factors associated with PHQ-9 scores in the subset of ART women, some significant factors were found in the whole group of participants: anxiety, perception of stressful life events, low resilience, low self-esteem, and poor partner support.

Pregnant women after ART had lower levels of quality of life, particularly in social and physical spheres than the women who conceived spontaneously [23]. The decrease in these vital spheres could be because ART carries a higher risk of obstetrical complications [57,58] requiring social and physical restriction. According to our results in the first trimester, we did not detect significant differences between depression score and type of reproduction (spontaneous/ART), although we posited that ART and depression symptomatology were associated, since up to 32% of women who conceived via of ART scored above the PHQ-9 cut-off. The literature describes that ART women felt higher levels of pregnancy-specific anxiety, regarding miscarriage and abortions, but the same or even lower depressive symptomatology than spontaneous pregnancy women [23]. Among the potential explanations for the discrepant results, it could be included that many of the studies explored did not control for confounding variables. Most studies did not clarify whether the psychological assessment was performed before or after the prenatal screening, an important variable to predict depression and anxiety during pregnancy [59]. Importantly, this study shows that, although at the beginning of pregnancy, they did not necessarily have a consolidated symptomatology of depression, they may have symptoms, which correlate with other variables to which health professionals need to pay attention as signs of alarm. Considering our results, PHQ-9 could be used to screen depression symptomatology in ART pregnancies at the beginning of gestation. The use of a screening tool to identify women at risk of depression during pregnancy should be universal practice to promote the wellbeing of mothers and infants [16].

Depressed women could have poorer self-care during pregnancy directly impacting obstetrical care and subsequently the health of the neonate [32]. Antenatal depression is associated with reduced maternal responsiveness to the infant [60]. To prevent depression feelings during pregnancy, increasing maternal coping could be effective [30]. According to our adjusted models, the PHQ-9 score was predicted by ART. These findings could support the proposal that once the women have been tested with the PHQ-9, they could benefit from psychological interventions, to reduce depression symptomatology, including reducing anxiety, handling stress live events and reinforcing maternal resilience and self-esteem, even more important in ART women. Certainly, maternal psychology should be considered along with ART, particularly important to determinate what standard instruments need to be used to avoid general norms and to develop precise depression screening [23].

Maternal age, partner support, employment, and socioeconomical status of the women could play a key role in depression feelings during pregnancy. These variables could be a key determinant to lead maternal depression. According to our data, although work status was not associated with depression score, advance maternal age and primiparous status were significantly different with ART, and single women and disrupted partner support were risk factors for increased PHQ-9 score. These characteristics may increase the coping demands set on the pregnant woman, thereby increasing the overload of general anxiety and closing the trigger of depression. Viewing the Lazarus and Folkman theory as an overall maternal mental health background, these factors could be perceived as negative emotional consequences, leading to depression symptomatology. Many studies found association between young age and depression and anxiety during pregnancy. Thus, adolescents are at increased risk of becoming depressed during pregnancy [31,61,62]. In addition, perceived partner support and marital satisfaction are protective factors against antenatal depression [63,64,65]. Furthermore, antenatal depression symptomatology seems to increase in unemployed women and housewives [64,66]. Moreover, the employee status of the partner seems to also be linked with maternal antenatal depression [67]. However, studies using the maternal economic status to predict depressive symptomatology during pregnancy showed contradictory results. Some studies showed that low financial income is a risk factor for maternal depression [68,69], and other studies did not find any association [67,70]. Although our data support the psychological independent effect of socioeconomical status on depression symptomatology, this result should be deeply explored because other variables such as ethnicity, previous history of miscarriage, time of infertility, or even educational level could be modulating the interaction, considering the homogeneity of our cohort. Considering our data and the literature, the socioeconomical characteristics could signify contextual conditioners to establish early psychological follow-up protocols for pregnant women, particularly important in ART. Anxiety may lead to depressive symptomatology without effective coping tools to ensure adequate channeling.

Within the psychological factors of depressive symptomatology, reduced self-esteem has been proposed as a predictor of depression during pregnancy [71]. Pregnant women with high levels of self-esteem and optimism have low perceived stress [72], which could be a protective factor against depression symptoms. Our data support this idea; during the first trimester, self-esteem was a protective factor against depression symptomatology. In addition, we found the same pattern with resilience and partner support. Interestingly and as a positive interpretation, some data showed that ART women had the same level of self-esteem and even more positive attitudes toward pregnancy than spontaneous pregnancy women [23]. If we focus on the self-esteem of pregnant women, we can strengthen resilience, which is a protective factor not only for antenatal but also for postpartum depression [73,74]. Additionally, another study has reported that maternal optimism at the end of pregnancy showed positive associations with maternal age, marital status, optimism, and resilience in the first trimester of gestation [30]. Considering marital status, we found a protective effect of partner support decreases PHQ-9 score. This reinforces the idea that early psychological intervention and following-up support during pregnancy are the key roles of the sociodemographic context. As a primary intervention, the empowerment of women in self-esteem and resilience could be critical to counteracting negative emotional consequences, particularly in women who conceive via ART.

Negative feelings can be mediated by stressful events during pregnancy, which can lead to depression. Stress may reduce the successful ART outcome, possibly through psychobiological mechanisms [75] by increase of cortisol and the dysregulated inflammatory profile of the women, which lead to placental hypoperfusion affecting fetal growth and neurodevelopment [27,76,77,78]. However, a weakness of studies evaluating maternal stressful events is their lack of specificity [27], which conditions the heterogeneity of the conclusions. In a Danish study of ART women that analyzed perceived stress by the list of recent events questionnaire, many perceived stressful life events had a negative impact on quality of life [75]. In immediately postpartum women from Spain, no association has been reported between anxiety and depression, and perceived stressful life events [29]. Considering our data in the first trimester and controlled by ART, the stressful life events checklist was associated with depression symptomatology. Contrary, the life stress category of PDPI-R was not associated with depression. It seems interesting to note at this point that the events explored in the checklist could be less anxiety inducing but more prevalent than those explored in the PDPI-R. In this context, our study provides relevant information on the frequency and intensity of stressful life events that should be assessed in early pregnancy. Therefore, they could become a central element in psychological screening of depression. Additionally, our GAD-7 measure predicted the PHQ-9 score, being anxiety an early risk factor for depression in ART pregnancies. Globally in Lazarus and Folkman’s theory [24] as applied to maternal mental health during pregnancy [79], anxiety could activate negative maternal feelings, leading to perceive situations as being more stressful than normal [80].

Therefore, according to our measurement of general concerns at the beginning of pregnancy, these were positively associated with depression symptomatology, although this was not significant in the model. Another Spanish study found that pregnancy concerns in the second trimester were associated with depression symptoms at the end of gestation [30]. Our data could show that general concerns are present at the beginning of pregnancy and that these concerns may become more pregnancy-specific and should be evaluated. Antenatal depression can be prevented with appropriate interventions and by increasing knowledge of the factors involved [81], which is particularly important in women with ART. A deeper understanding of the protective and risk factors for depression during gestation will ensure a more humane pregnancy.

### Limitations and Future Directions

Sociodemographic characteristics were homogeneous, which could circumscribe the conclusion to a particular population. However, the homogeneity can be favorable for interpreting the conclusions in terms of psychological variables between types of reproduction because there were not socio-modulatory variables that could uncover the found associations.

Secondly, the dependent and predictive variables were measured in the same timeframe, potentially affecting the found interrelationship. Although, the enrolling effort, the sample size, and exclusion criterion could solve this issue, it would be desirable to perform new longitudinal studies to explore deeply the contribution of risk/protective factors associated with depression scores in pregnant women.

On the other hand, it is important to highlight that the interesting and relevant aspect of this article was that variables associated with depressive symptomatology can be isolated early in the pregnancy, particularly in women who conceived via ART, which is becoming increasingly prevalent in industrialized countries. This means that from a psychological point of view, early screening can be done, and coping strategies can be reinforced to empower the women in regard to positive variables and minimize the effects of negative variables to prevent long-term depressive symptomatology. However, it would be important to consider that the sample size of women derived from ART was small. It would be desirable to expand the women enrollment with this type of gestation to corroborate our findings.

## 5. Conclusions

Overall, we can conclude that advanced maternal age and primiparous status were factors associated with ART. During early pregnancy, anxiety, stressful life events, resilience, self-esteem, social support, and socioeconomic status could be intervention areas related to depression symptoms, even more important in single women whose pregnancy was derived by ART. Empowerment of ART women with psychological tools can reduce anxiety, stressful life events, and general concerns, and improving their resilience, self-esteem, and emotional partner support can be beneficial for preventing depression symptomatology.

This work provides insights for psychological adjustment in ART pregnancy. Health-care professionals should enhance the implementation of psychological screenings for high-risk pregnant women. They should provide counseling for depressed pregnant women. Adequate training of obstetricians and midwives in the detection and management of depression among pregnant women should help to decrease the psychological burden during pregnancy.

## Figures and Tables

**Figure 1 jcm-10-05672-f001:**
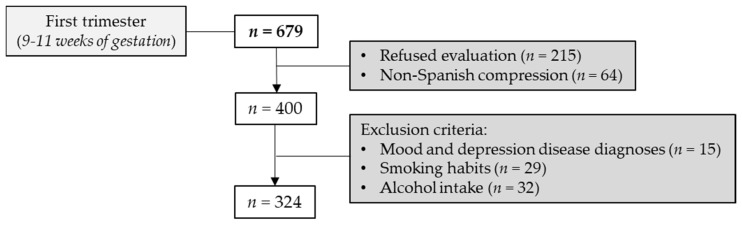
Flow chart of study participants. Sample size (*n*) is shown between brackets.

**Table 1 jcm-10-05672-t001:** Sociodemographic characteristics compared to assisted reproduction techniques.

		Assisted Reproduction Techniques (ART)		*p*-Value
	Total (*n* = 324)	No (*n* = 291)	Yes (*n* = 30)	
Maternal age (years)	34.0 [30.0; 36.0]	34.0 [30.0; 36.0]	35.0 [32.0; 38.0]	0.028
Marital status				0.741
Married	212 (65.4%)	191 (65.6%)	21 (70.0%)	
Separated	1 (0.3%)	1 (0.3%)	0 (0.0%)	
Single	108 (33.3%)	99 (34.1%)	9 (30.0%)	
Education				0.887
Primary degree	148 (45.7%)	135 (46.4%)	13 (43.3%)	
High School degree	3 (0.9%)	3 (1.0%)	0 (0.0%)	
University degree	170 (52.5%)	153 (52.6%)	17 (56.7%)	
Country of birth				0.313
Spain	292 (90.1%)	266 (91.4%)	26 (86.7%)	
Europe	8 (2.5%)	7 (2.4%)	1 (3.3%)	
Latin America	15 (4.6%)	12 (4.1%)	3 (10.0%)	
Other	6 (1.9%)	6 (2.1%)	0 (0.0%)	
Employment				0.318
Employed	238 (73.7%)	213 (73.2%)	25 (83.3%)	
Unemployed	83 (25.6%)	78 (26.9%)	5 (16.7%)	
Comorbidities	54 (16.7%)	48 (16.6%)	6 (20.0%)	0.823
First time of maternity	153 (47.2%)	133 (45.7%)	20 (66.7%)	0.046
Abortions before	57 (18.8%)	52 (19.2%)	5 (17.2%)	0.996
C-section before	24 (7.5%)	23 (8.0%)	1 (3.3%)	0.713

Data show median and interquartile range [Q1; Q3] in quantitative variables and sample size (*n*) and relative frequency (%) in qualitative variables. The *p*-Values were extracted using a Mann–Whitney’s U test with quantitative variables and a Fisher’s exact test with qualitative variables. The *p*-Value was compared between ART and non-ART groups.

**Table 2 jcm-10-05672-t002:** Sociodemographic characteristics compared to depression scale.

	PHQ-9		*p*-Value
	<10 (*n* = 260)	≥10 (*n* = 49)	
Maternal age (years)	33.0 [30.0; 36.0]	35.0 [31.0; 36.0]	0.218
Marital status			0.193
Married	176 (68.0%)	28 (57.1%)	
Separated	0 (0.0%)	1 (2.0%)	
Single	84 (32.3%)	20 (40.8%)	
Education			0.088
Primary degree	118 (45.4%)	23 (46.9%)	
High School degree	1 (0.4%)	2 (4.1%)	
University degree	141 (54.2%)	24 (49.0%)	
Country of birth			0.475
Spain	235 (90.4%)	46 (93.9%)	
Europe	9 (3.5%)	0 (0.0%)	
Latin America	10 (3.9%)	3 (6.1%)	
Other	6 (2.3%)	0 (0.0%)	
Employment			0.997
Employed	191 (73.5%)	35 (72.9%)	
Unemployed	69 (26.5%)	14 (28.6%)	
Comorbidities	41 (15.8%)	10 (20.4%)	0.561
First time of maternity	131 (50.4%)	19 (38.8%)	0.182
Abortions before	45 (18.4%)	7 (15.9%)	0.859
C-section before	17 (6.6%)	4 (8.2%)	0.757
ART	19 (67.9%)	9 (32.1%)	0.030

Data show median and interquartile range [Q1; Q3] in quantitative variables and sample size (*n*) and relative frequency (%) in qualitative variables. Patient Health Questionnaire (PHQ-9); Assisted reproduction techniques (ART). The *p*-Values were extracted using a Mann–Whitney’s U test with quantitative variables and a Fisher’s exact test with qualitative variables.

**Table 3 jcm-10-05672-t003:** Correlations of the psychological variables with depression scale.

	Rho	*p*-Value		Rho	*p*-Value
GAD-7	0.54	<0.001	PDPI-R-Self-esteem	−0.26	<0.001
Resilience	−0.33	<0.001	PDPI-R-Partner support	−0.16	0.004
General concerns	0.22	<0.001	PDPI-R-Family support	−0.18	0.002
SLE	0.35	<0.001	PDPI-R-Friends support	−0.20	<0.001
PDPI-R-SES	−0.14	0.012	PDPI-R-Marital satisfaction	0.03	0.572
PDPI-R-Pregnancy intendedness	−0.08	0.173	PDPI-R-Life stress	0.17	0.003

Data show Spearman’s rho coefficients and *p*-value associated. Generalized Anxiety Disorder 7-item Scale (GAD-7), Patient Health Questionnaire (PHQ-9), Stressful Life Events (SLE), Postpartum Depression Predictor Inventory-Revised (PDPI-R), Socioeconomical Status (SES). The *p*-value < 0.05 was considered significant.

**Table 4 jcm-10-05672-t004:** Generalized regression models associated with PHQ-9 scale.

Model 1	β ± SE	*p*-Value	Model 2	β ± SE	*p*-Value	Model 3	β ± SE	*p*-Value
ART	6.75 ± 0.74	<0.001	ART	9.60 ± 2.26	<0.001	ART	16.51 ± 3.22	<0.001
			Maternal age	−0.07 ± 0.06	0.211	Marital status (Ref. Married)	0.74 ± 0.41	0.072
			Marital status (Ref. Married)	1.14 ± 0.50	0.022	GAD−7	0.43± 0.06	<0.001
			Education (Ref. University)	0.10 ± 0.49	0.845	Resilience	−0.05 ± 0.02	0.004
			Country of birth (Ref. Spain)	−1.05 ± 0.81	0.846	Stressful life events	0.17 ± 0.06	0.003
			Employment (Ref. Employed)	0.35 ± 0.55	0.518	General concerns	0.12 ± 0.07	0.074
			Comorbidities	0.09 ± 0.62	0.881	PDPI-R-Socioeconomic status	0.03 ± 0.36	0.931
			First time of maternity	−0.67 ± 0.54	0.215	PDPI-R-Self-esteem	−1.21 ± 0.61	0.048
			Abortions before	−0.98 ± 0.66	0.139	PDPI-R-Partner support	−1.50 ± 0.60	0.013
			C-section	0.18 ± 1.02	0.863	PDPI-R-Family support	−0.23 ± 0.39	0.556
						PDPI-R-Friends support	0.01 ± 0.27	0.963
						PDPI-R-Life stress	0.06 ± 0.21	0.777
R^2^ = 0.02			R^2^ = 0.20			R^2^ = 0.60		

Data show estimated beta coefficients (β) ± standardized error (SE). Assisted reproduction techniques (ART), Reference (Ref.), Generalized Anxiety Disorder 7-item Scale (GAD-7), Patient Health Questionnaire (PHQ-9), Postpartum Depression Predictor Inventory-Revised (PDPI-R). Adjusted R-squared (Adj. R^2^).

**Table 5 jcm-10-05672-t005:** Generalized regression models associated with PHQ-9 scale in the subset of ART women.

Model in ART Subset	β ± SE	*p*-Value
Marital status (Ref. Married)	−0.66 ± 2.67	0.810
Resilience	−0.02 ± 0.11	0.840
Stressful life events	0.30 ± 0.25	0.258
General concerns	−0.03 ± 0.38	0.939
PDPI-R-Socioeconomic status	−2.80 ± 2.50	0.288
PDPI-R-Self-esteem	1.79 ± 4.71	0.713
PDPI-R-Partner support	−3.51 ± 3.51	0.341
PDPI-R-Family support	−3.01 ± 2.75	0.299
PDPI-R-Friends support	2.46 ± 2.03	0.253
PDPI-R-Life stress	−0.27 ± 1.50	0.863
R^2^ = 0.35		

Data show estimated beta coefficients (β) ± standardized error (SE). Reference (Ref.), Patient Health Questionnaire (PHQ-9), Postpartum Depression Predictor Inventory-Revised (PDPI-R). Adjusted R-squared (Adj. R^2^).

## Data Availability

The data presented in this study are available on request from the corresponding author. The availability of the data is restricted to investigators based in academic institutions.

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
