# Peer review of "Early Association Factors for Depression Symptoms in Pregnancy: A Comparison between Spanish Women Spontaneously Gestation and with Assisted Reproduction Techniques"

_jcm, 2021, doi:10.3390/jcm10235672_

Round 1
Reviewer 1 Report
The manuscript entitled "Early risk and protective factors to depression symptoms in women with assisted reproduction techniques" aimed to explore the association of ART use with antenatal depression by using the cross-sectional study. The theme is important for perinatal mental health. However, several methodological concerns have remained.
Major comments.
Introduction
1. (P.1 LN43)Why authors thought ART was potentially associated with antenatal depression is unclear. A sufficient explanation is needed based on any previous model and theory.
Methods and Results
2. Curiously, overall PHQ-9 scores were not different between women with ART and non-ART, but the crude model of regression analysis showed a significant increase among women with ART. Normally the same result will be obtained.
3. Adjusted model 3 was inappropriate. If you explore the risk and protective psychological factors for depression among women with ART, please conduct a stratified analysis limiting the sample to women with ART. In addition, anxiety (GAD-7) is rather an outcome than the associated factor for depression. I recommend authors exclude this variable. Please also consider including sociodemographic characteristics at the same time in model 3.
Discussion
4. The first paragraph of the discussion should be mentioned the main findings of this study. The discussion section had much unnecessary information and potential readers may be confused.
5. Please keep focusing on the PECO of this study. The paragraph regarding maternal complications and PHQ-9 scale utility may be out of your scope. If the authors would like to mention it, please include this topic also in the introduction.
6. P.8 LN27, authors stated that significant differences were not detected. Please keep consistent with the abstract. The abstract is misleading. Please be modest to the main findings.
Minor comments.
1. Please consider including "cross-sectional study" in the title, following the ICMJE recommendations.
Introduction
2. (P.2, LN84-87)Literature 24 and 25 were not the studies for women with ART. The phrase in the next sentence "especially in women with ART" was misleading. Is there any study to investigate associated factors for mental health among women with ART? If no, please say there is no evidence.
Methods
3. (P.3 LN10) Please add more information about the hospital to ensure the generalizability of the study findings.
4. Did the authors invite "all" pregnant women to visit the hospital? Recruiting time (2014-2015) was one year (right?). I wonder if some selected pregnant women were invited to participate in the study. It would be a crucial bias.
5. Please add detailed information on covariates in the regression model in the statistical analysis section.
Results
6. (Table 1) The total of No (n=291) and Yes (n=30) is not total (n=324).
7. (P.6 LN23) Please include the result of the women with PHQ-9>10 was high in ART in Table 2.
Author Response
The manuscript entitled "Early risk and protective factors to depression symptoms in women with assisted reproduction techniques" aimed to explore the association of ART use with antenatal depression by using the cross-sectional study. The theme is important for perinatal mental health. However, several methodological concerns have remained.
Response: Thank you for your time reviewing our article. Please see below our point-by-point responses.
Major comments.
Introduction
- (P.1 LN43) Why authors thought ART was potentially associated with antenatal depression is unclear. A sufficient explanation is needed based on any previous model and theory.
Response: Thank you to make us think in this point. There are studies that find women who conceive by ART have a qualitatively different experience of pregnancy. However, there are controversy about this. In the other hand, there is no postulated model which explain why depression during pregnancy could be associated with ART, but there are described factors that could induce it in these women. According to our thoughts, it could be determinate the moment when depression symptoms and its associated factors were measurement. Furthermore, most of the works were focus on postpartum depression, which it would be different of antenatal depression. This information was extended in the text (lines 45-52).
Methods and Results
- Curiously, overall PHQ-9 scores were not different between women with ART and non-ART, but the crude model of regression analysis showed a significant increase among women with ART. Normally the same result will be obtained.
Response: This is a very good observation. We used the rank test (non-parametric) to determine whether there were differences between the groups. This test checks whether the ranks between two factors were equal (in our case, the sum ranks were equal). However, women with ART scored higher than women with spontaneous pregnancies (in median, even more if we would use the average non-ART=5.6 versus ART=6.8). On the other hand, generalized regression models predict the score of one variable (in our case PHQ-9 scores as a dependent variable) as a function of another (presence or absence of ART). It determines whether the coefficient by each factor is significant in determining the score of the dependent variable. This was significant in our unadjusted model.
- Adjusted model 3 was inappropriate. If you explore the risk and protective psychological factors for depression among women with ART, please conduct a stratified analysis limiting the sample to women with ART. In addition, anxiety (GAD-7) is rather an outcome than the associated factor for depression. I recommend authors exclude this variable. Please also consider including sociodemographic characteristics at the same time in model 3.
Response: We appreciate these suggestions. We decided to include the entire cohort in model 3 because the sample size of the ART group was small, which was included as limitations of the study (lines 446-449). Moreover, including the anxiety in the model to predict depression scores could make the model adjust for this variable as well. Furthermore, there are evidence that suggest the anxiety as a risk factor for depression by itself (PMID: 15234824).
In the other hand, initially the sociodemographic variables were not included in model 3 as they did not show significance in model 2. But we have considered this comment very valuable and have applied model 3 with the significant variable of model 2. In addition, as suggested by the reviewer, a complementary model was added as supplemental material S2 for the interest of the readers.
Discussion
- The first paragraph of the discussion should be mentioned the main findings of this study. The discussion section had much unnecessary information and potential readers may be confused.
Response: We have added a general summary at the beginning of the discussion section (lines 316-318). Also, we have summarized some parts to make it more focused. In closing, we believe that this revised manuscript now addresses more specific comments.
- Please keep focusing on the PECO of this study. The paragraph regarding maternal complications and PHQ-9 scale utility may be out of your scope. If the authors would like to mention it, please include this topic also in the introduction.
Response: We appreciate your interest in our work and believe that the reviewers’ comments have greatly improved the quality of the manuscript. As reviewer mentioned, we have removed the paragraph from the discussion to make it more readable.
- 8 LN27, authors stated that significant differences were not detected. Please keep consistent with the abstract. The abstract is misleading. Please be modest to the main findings.
Response: We apologize for any inconvenience this type of error may have caused. We have carefully revised the text to match with the abstract. Thank you again for your time and consideration.
Minor comments.
- Please consider including "cross-sectional study" in the title, following the ICMJE recommendations.
Response: We have considered the reviewer´s suggestions and we agree and have added this term in the title.
Introduction
- (P.2, LN84-87). Literature 24 and 25 were not the studies for women with ART. The phrase in the next sentence "especially in women with ART" was misleading. Is there any study to investigate associated factors for mental health among women with ART? If no, please say there is no evidence.
Response: Considering the reviewer´s suggestion, we have reformulated the text to do more understandable.
Methods
- (P.3 LN10) Please add more information about the hospital to ensure the generalizability of the study findings.
Response: We have updated the information about hospital of recruitment (lines 110-112).
- Did the authors invite "all" pregnant women to visit the hospital? Recruiting time (2014-2015) was one year (right?). I wonder if some selected pregnant women were invited to participate in the study. It would be a crucial bias.
Response: Considering the reviewer´s comments, we have modified the text to do more understandable the recruitment and study design (lines 112-117).
- Please add detailed information on covariates in the regression model in the statistical analysis section.
Response: We appreciate this comment. Considering a covariate as any continuous variable, which were measured but not randomization was controlled in the data collection. All variables that were introduced in the models were covariables, and the standardized beta coefficients were adjusted by all input variables introduced.
Results
- (Table 1) The total of No (n=291) and Yes (n=30) is not total (n=324).
Response: Thank you for this appreciation. There were 3 cases that we did not have the information about their type of gestation.
- (P.6 LN23) Please include the result of the women with PHQ-9>10 was high in ART in Table 2.
Response: The table 2 was updated according to the reviewer's comment.
Reviewer 2 Report
Thank you for giving me the opportunity to review this manuscript.
It is necessary to revise this manuscript before publication.
1) Please attach STROBE checklist, and add some sentences based on the statement. For example, please clarify the study design in the abstract and the method section. Please make it clear what is the PECO in the study, what is confounders, and what is dependent variables. Please explain the primary and secondary outcomes of this study (to avoid type 1 errors). Furthermore, please show sample size estimation especially in pregnant women with ART. Please show potential risk of bias in this study.
2) I have the following concerns in the result and discussion section.
a) In the Model 1, the authors showed that ART is associated with depression, but is the results shown as an unadjusted model? In the discussion, the authors explained that "we posit ART and depression symptomatology was associated, since up to 32% (N = 9-10) of women who conceived via of ART scored above the PHQ-9 cut-off. ", but it is difficult to make this conclusion. In this regard, please revise "in Women with Assisted Reproduction Techniques" as "in pregnant women: Findings from Spain" in the title.
b) In the Model 2, the authors showed “Behavior of Psychological variables with Depression Scale”. In this analysis, was the target population pregnant women both with ART and without ART? Was this analyzed to find potential confounders of the association between ART an depression? Please explain the aim of this analysis, so that psychological and demographic variables would be adjusted in the model 3.
c) In the Model 3, the authors showed "Depression Scale and ART models Controlled by Demographic and Psychological variables". What kind of Demographic and Psychological variables controlled to find the association between ART and depression in pregnant women, because it is still unclear to demonstrate the association of ART and depression by the authors' explanation? Did authors include the GAD, PDPI, Resilience, general concerns, SLE, and life stress as the confouders (found in the model 2) in the model 3?
I think It is necessary to revise the manuscript before publication because of the reasons above.
Author Response
Thank you for giving me the opportunity to review this manuscript. It is necessary to revise this manuscript before publication.
Response: Thank you for your time reviewing our article. Please see below our point-by-point responses.
- Please attach STROBE checklist, and add some sentences based on the statement. For example, please clarify the study design in the abstract and the method section. Please make it clear what is the PECO in the study, what is confounders, and what is dependent variables. Please explain the primary and secondary outcomes of this study (to avoid type 1 errors). Furthermore, please show sample size estimation especially in pregnant women with ART. Please show potential risk of bias in this study.
Response: Thank you for this value suggestions. STROBE checklist was added as a table S1, and we have modified the material and methods and abstract. In addition, the statistical and introduction sections were modified to match with PECO statements (PMID: 30166065). We have included the sample size calculation according to our dependent variable (lines 107-109) to avoid potential risk bias in the study. We believe that the reviewers’ comments have greatly improved the quality of the manuscript.
- I have the following concerns in the result and discussion section.
- In the Model 1, the authors showed that ART is associated with depression, but is the results shown as an unadjusted model? In the discussion, the authors explained that "we posit ART and depression symptomatology was associated, since up to 32% (N = 9-10) of women who conceived via of ART scored above the PHQ-9 cut-off", but it is difficult to make this conclusion. In this regard, please revise "in Women with Assisted Reproduction Techniques" as "in pregnant women: Findings from Spain" in the title.
Response: We want to thank the reviewer to make us think in these aspects. Below we summarize the revisions made in response to the reviewers’ suggestions. As the reviewer points out, model 1 is raw or unadjusted. In addition, the sample size of the women in the ART group was limited, and we have included this aspect in the limitations of the study (lines 446-449). Otherwise, in the discussion we established a suggestion and was not stated as a conclusion of the study. Qualitatively, pregnant women derived from ART have a different psychological pattern from women who with spontaneously pregnancy. Therefore, there are risk and protective factors that could keep them away from antenatal depressive symptomatology, particularly in ART. We have deeply considered the modification in the title, but many of the data were analyzed from the point of view of ART. In addition, Spain could be considered as a one of the European countries with high rate of ART, even it has been called by social media as the country of “reproduction tourism” (https://www.tourism-review.com/fertility-tourism-developing-fast-in-spain-news10817).
- In the Model 2, the authors showed “Behavior of Psychological variables with Depression Scale”. In this analysis, was the target population pregnant women both with ART and without ART? Was this analyzed to find potential confounders of the association between ART and depression? Please explain the aim of this analysis, so that psychological and demographic variables would be adjusted in the model 3.
Response: Model 2 is not exactly the “Behavior of Psychological variables with Depression Scale” rather than would be the section 2. In section 2, we explored how the psychological variables correlate with the dependent variable (PHQ-9 scores). This analysis was done on the full cohort and guide us to choose what the psychological variables are the real candidates to be introduced in the posterior model (model 3) that explains the variance of the PHQ-9 scores. This has been clarified in the text to avoid confusion (lines 279-280).
- In the Model 3, the authors showed "Depression Scale and ART models Controlled by Demographic and Psychological variables". What kind of Demographic and Psychological variables controlled to find the association between ART and depression in pregnant women, because it is still unclear to demonstrate the association of ART and depression by the authors' explanation? Did authors include the GAD, PDPI, Resilience, general concerns, SLE, and life stress as the confouders (found in the model 2) in the model 3?
Response: In the models, we used the "intro" method, therefore, all the standardized beta coefficients were adjusted by all input variables introduced. It can be considered that they were not modulating rather than covariables. Since our interest was in determining how each variable predicted (associated) the PHQ-9 scores in the presence of the variable “type of gestation” (ART/non-ART, considering non-ART as the reference). Section 2 allowed us to clarify which variables would be included in model 3.
I think it is necessary to revise the manuscript before publication because of the reasons above.
Response: We hope that we have adequately satisfied your comments. Your suggestions have improved the article considerably.
Round 2
Reviewer 1 Report
Thank you for revising the manuscript. I think it was greatly improved. But several points are still needed to be considered.
1. (Table S2.) There was no data inputted in PDPI-R-Life stress. Please check it.
Introduction
2. Authors stated that "The purpose of this study as the primary outcome was to determine what sociodemographic of factors and psychological variables could impact on early depression symptoms in pregnant women undergoing ART.". If so, Table S2 is the primary finding and is more important. This paper examined:
(i) Is ART associated with depression?
(ii) What sociodemographic and psychological factors were associated with depression in women with ART?
(iii) What sociodemographic and psychological factors were associated with depression in pregnant women?
Please clearly describe the study objectives in the last paragraph of the introduction.
Methods
3. LN 112 "A total of 679 women were contacted during her first trimester of pregnancy."
How were they selected? All the women during the first trimester visiting the hospital were invited? Please add more information on how to select the potential participants.
Discussion
4. The summary findings of this study in Lines 316-318 were still not enough. Please answer your PECO clearly in the first sentence and separate the sentences related to the associated factors for depression in women with ART. I think the psychological factors related to depression were "sub" findings. Stratified analysis (Table S2) showed that this study did not find any associated factors in women with ART. Please consider the expressions below, for example,
"In this study, we found that in early pregnancy, scores of the PHQ-9 scale were associated with pregnant women derived by ART. While there were no significant psychological factors associated with PHQ-9 scores in the subset of ART women, some significant factors were found in the whole participants: anxiety, perception of stressful life events, low resilience, low self-esteem, and poor partner support."
Conclusion
5. The sentences related to PHQ-9 utility should be excluded.
Author Response
Thank you for revising the manuscript. I think it was greatly improved. But several points are still needed to be considered.
- (Table S2.) There was no data inputted in PDPI-R-Life stress. Please check it.
Response: Our sincere apologies for this typographical error. The table has been updated. Thank you for your comment.
Introduction
- Authors stated that "The purpose of this study as the primary outcome was to determine what sociodemographic of factors and psychological variables could impact on early depression symptoms in pregnant women undergoing ART.". If so, Table S2 is the primary finding and is more important. This paper examined:
- Is ART associated with depression?
- What sociodemographic and psychological factors were associated with depression in women with ART?
- What sociodemographic and psychological factors were associated with depression in pregnant women?
Please clearly describe the study objectives in the last paragraph of the introduction.
Response: We appreciate your interest in our work and believe that the reviewer’s comment has greatly improved the quality of the manuscript. According to the suggestions, the table S2 was incorporated in the main text as a table 5.
In addition, these were good questions. There is much controversy as to whether ART is associated with depression, considering that there are no models/theories, we consider that variables such as timing of measurement during pregnancy, maternal age, years of infertility or previous abortion experiences may play a decisive role. We have included this information in the text. Furthermore, some of the social variables described and associated with depression in ART were unemployment, low educational level, or socioeconomic status. In the case of psychological variables in ART with depression. There are few studies that explore this in this group of women. If it is known that ART is associated with high perception of stressful life events, anxiety, general concerns, or low self-esteem. And these are variables that could increase depression symptomatology. We have included these variables throughout the introduction. In the other hand, the aims were specifically described (lines 98-108).
Methods
- LN 112 "A total of 679 women were contacted during her first trimester of pregnancy." How were they selected? All the women during the first trimester visiting the hospital were invited? Please add more information on how to select the potential participants.
Response: Thank you very much for your appreciation, it has helped us to better understand the recruitment. We have modified the text included (lines 117-119).
Discussion
- The summary findings of this study in Lines 316-318 were still not enough. Please answer your PECO clearly in the first sentence and separate the sentences related to the associated factors for depression in women with ART. I think the psychological factors related to depression were "sub" findings. Stratified analysis (Table S2) showed that this study did not find any associated factors in women with ART. Please consider the expressions below, for example, "In this study, we found that in early pregnancy, scores of the PHQ-9 scale were associated with pregnant women derived by ART. While there were no significant psychological factors associated with PHQ-9 scores in the subset of ART women, some significant factors were found in the whole participants: anxiety, perception of stressful life events, low resilience, low self-esteem, and poor partner support."
Response: This was a great help, and we thank you very much for your time and dedication in reviewing our work. We believe that the evaluation fits the results as well as the target audience. We have included table S2 as table 5 in the main text and re-written the beginning of the discussion to fit the content of the article, as commented by the reviewer.
Conclusion
- The sentences related to PHQ-9 utility should be excluded.
Response: Thank you for your time dedicated to our work. We have removed this sentence to make the work more focused.
Reviewer 2 Report
I belive that this manuscript has been sufficiently imrpve.
Author Response
I believe that this manuscript has been sufficiently improve.
Response: Thank you very much for taking the time to review our work. It has been a great help to improve it.